# Anthropogenic nutrient inputs cause excessive algal growth for nearly half the world's population

Richard W. McDowell [1,2] ✉, Dongwen Luo[3], Peter Pletnyakov[1], Martin Upsdell[3] & Walter K. Dodds [4]

Reference conditions pertain to conditions without anthropogenic influence and serve to gauge the degree of river pollution and identify the best attainable water quality. Here we show estimates of the global human footprint of nitrogen and phosphorus concentrations and potential for related nuisance or harmful algal growth in rivers. We use statistical models based on 1.2 million stream nutrient measurements (from 2005 to 2013) and find global human enrichment of river total nitrogen and total phosphorus is 35% and 14% respectively. The greatest enrichment is in Europe (86 and 30% respectively) and the least in Oceania (9 and 2% respectively). The levels of enrichment translated into an almost doubling of the catchment areas with rivers predicted to have anthropogenically elevated levels of potentially harmful or nuisance algae, affecting ~40% of the world's population. Focusing management on the difference between current and reference conditions can help protect good water quality while avoiding unrealistic goals where nitrogen and phosphorus are naturally high.

Enriched nitrogen and phosphorus in rivers and streams can pollute waters and stimulate autotrophic and heterotrophic states[1]. Stimulation of autotrophic state often leads to elevated levels of algal biomass, particularly attached algae (periphyton) that can be toxic, lead to taste and odour problems, deplete dissolved oxygen, interfere with recreation, harm biodiversity, and cause objectionable appearance[2–5]. Many regulatory bodies and jurisdictions rely on the concept of reference conditions as the best attainable state of ecosystems. These reference conditions typically denote the concentration or yield of nitrogen and phosphorus in rivers that would occur with negligible human influence or disturbance[6] and serve as a benchmark to gauge how perturbed the river is and determine the potential for improvement[7]. Knowledge of reference conditions also avoids the setting of catchment objectives that may be unachievable (e.g., when desired nutrient concentrations are below the reference

level, so regular methods of nutrient control through land management could not reach the desired concentrations).

Reference conditions have largely been defined in developed countries (defined as having a gross national income per capita >14,005 United States [US] dollars in 2024 by the World Bank) with abundant data and information on river and catchment characteristics. These characteristics have been used to extend reference conditions across classes of similar biophysical parameters like climate, topography, and geology[8], thus avoiding irrelevant classifications based on geopolitical boundaries. For example, in the US, reference conditions are classed according to 14 ecoregions that span multiple States[9]. In the European Union, the Water Framework Directive uses reference conditions to inform thresholds for 'good' ecological status and intercalibrates these across member countries[10]. Biophysical classes recognise that nitrogen and phosphorus losses will vary among sites according to reference conditions as influenced by factors such as

[1]AgResearch, Lincoln Science Centre, Christchurch, New Zealand. [2]Faculty of Agriculture and Life Sciences, Lincoln University, Lincoln, Christchurch, New Zealand. [3]AgResearch, Ruakura Research Centre, Hamilton, New Zealand. [4]Division of Biology, Kansas State University, Manhattan, KS, USA. ✉ e-mail: Richard.mcdowell@agresearch.co.nz

climate, topography, and geology[11]. However, outside of developed nations the analysis and setting of reference conditions is rare[12–14].

Various methods are used to estimate reference conditions. The simplest are empirical, using existing sites under "minimally disturbed conditions" (MDC)[15]. The MDC approach defines reference conditions as those observations from rivers minimally affected by anthropogenic activities (e.g., <5% agricultural or urban land use and not receiving wastewater). However, such sites can be scarce, especially in regions with a long history of agriculture, and may represent only a few catchments. Moreover, rivers may be very sensitive to relatively low levels of urbanization because of lack of sewage treatment and no phosphorus detergent bans in some countries[16]. Another approach involves utilizing data from periods before a stream became degraded, known as "historical" reference conditions, but in many cases anthropogenic impacts pre-date the earliest water quality analyses. Lastly, the "least disturbed condition" approach groups sample data for sites and sets a reference condition at some subjective quantile or percentile (e.g., 5th percentile) at the lower end of the value's distribution but carries the risk of setting the reference condition too high if too many degraded sites are included or too low if many reference sites are in the dataset[17]. Other approaches combine water quality data from monitoring sites with catchment characteristics in mechanistic or stochastic models to estimate reference conditions. These methods benefit from a larger data pool and potentially produce outputs that are less biased and more representative of local conditions. Mechanistic approaches involve models like the Soil Water Assessment Tool that describe processes within the catchment and can be manipulated to output estimated reference conditions by setting the variables that capture anthropogenic influence to nil[18,19]. These models can provide accurate estimates of reference conditions but typically demand substantial fine-grained data, which may be lacking in many regions, and need to be fine-tuned for each catchment. Stochastic approaches group sites based on classifications that explain variation between sites and estimate reference conditions by setting anthropogenic variables (e.g., urban or cropland cover within the catchment above each sampling sites) within those classes to nil[8,9]. Stochastic models are favoured for their adaptability and ability to cover extensive areas but rely on the availability of sites within classes to make predictions across nutrient ecoregions.

In our work we utilise recently available global databases for nitrogen and phosphorus loads[20,21] and the likelihood of periphyton accrual (based on median nutrient concentrations[22]) to predict global references concentrations of both nitrogen and phosphorus forms in streams. We use an approach that is consistent globally and does not rely primarily upon data from developed countries. Then we compare these estimates with current nitrogen and phosphorus concentrations to assess the extent of anthropogenic influence on current nutrient concentrations. Finally, we predict standing stocks of periphyton under reference and current anthropogenic influence (in baseflow during warmer months) to indicate the degree to which eutrophication has harmed global rivers. These predictions can identify the anthropogenic influences on nutrient pollution and to inform policy to target actions to improve water quality on a global scale. They can also inform efforts to protect downstream waters, such as lakes and marine systems, which are also subject to problems of eutrophication[23].

## Results

### Controls on nitrogen and phosphorus losses under current and reference conditions

We used ~1.2 million measurements of nitrate-nitrogen, total nitrogen, dissolved reactive phosphorus and total phosphorus for up to 2618 catchments to generate models to predict median concentrations during baseflow for months likely to see periphyton growth (May – Oct or Nov-Apr for temperate zones in the Northern and Southern hemispheres or annually in the tropics). Additional information is available on the data process (Supplementary Fig. 1), modelling process (Supplementary Notes 1–3) and performance (Supplementary Notes 4 and 5), and outputs (Supplementary Notes 6–8). Our predictions for 14,241 catchments at level 6 of HydroBasins[15] performed well, as classed by ref. 24 based on the root mean square error and coefficient of determination. Post-processing, we noted a small amount (<1%) of total nitrogen data with low concentrations that could be considered as outliers. We chose not to remove them owing to their small number and close fit to the 1:1 line (testing data; see Supplementary Note 3).

Amongst the models, anthropogenic (i.e., 'human') effects (see also Supplementary Note 1 and Supplementary Fig. 2), accounting for land use, soil fertility, and population density, were 3–5 times more important than the next most important predictor for all nutrient forms except dissolved reactive phosphorus where human effects were 1.5 times more important. Population density and soil fertility (represented by Olsen phosphorus) were the most important components of the human effect predictor (see Supplementary Fig. 1). The remaining predictors included a combination of known drivers of nutrient loss[25] such as: potential evapotranspiration, runoff, and slope, but also latitude which may be acting as an amalgam of these predictors or to signal a seasonal effect (i.e., latitude was used to exclude data from colder months). Notably, slope was less important for nitrate-nitrogen than other nutrient forms given the lower adsorption of nitrate to soils relative to other forms of nitrogen and phosphorus. Our analysis indicated a strong influence of intensively farmed land, enriched soil fertility, and greater population density (especially on flat land) on increased river nutrient concentrations, as have others[16,26]. In many statistical approaches a dominant factor (like percentage cropland) is used to isolate the human effect[9]. Our method takes this a step further and accounts for multiple factors that could accrue to cause a human effect (e.g., total phosphorus is generally transported as sediment so slope of farmland can influence loads to rivers). We therefore combined these factors while avoiding any potential effect from multi-collinearity amongst variables (e.g., land uses).

We isolated sites under MDC in our dataset (*n* = 615) as the closest proxy for reference conditions available. MDC were defined as: Olsen P concentrations of ≤5 mg kg$^{-1}$, population density of ≤0.001 persons km$^{-2}$ and ≥80% forestland and no cropland or urban land (see also Supplementary Note 5). To standardise variation in biophysical conditions we contrasted the median values for sites under MDC to the predicted median reference values for terrestrial biomes as this would allow us to predict macroscale (up to global) variation in freshwater biogeochemistry[27]. Some analytes had few sites in each biome under MDC and more likely to be skewed than data-rich biomes. Of those with >5 observations, all were within the 95% confidence intervals for dissolved reactive phosphorus (*n* = 7), total phosphorus (*n* = 7), and nitrate-nitrogen (*n* = 6), while only one out of six biomes fell outside of the 95% confidence intervals for total nitrogen (Fig. 1). These contrasts suggest that reference conditions were well predicted by the model. We recognise that this is not a truly independent comparison (i.e., these sites are also included in the original model) but was necessary owing to the paucity of sites under MDC. We also compared our predicted values to estimates of published reference conditions for least disturbed conditions, MDC, and statistical approaches. While much of the published data exists for catchments that do not overlap with biomes or our catchments (i.e., less studied areas), reference values for total nitrogen and total phosphorus produced by MDC or statistical methods were in the same range as those predicted across 14,241 catchments globally (Fig. 2). Too few published values were available to make the same comparison for dissolved reactive phosphorus or nitrate-nitrogen.

Our outputs for catchments at level 6 of HydroBasins (*n* = 14,241) were derived from observations from 1186 to 2618 catchments

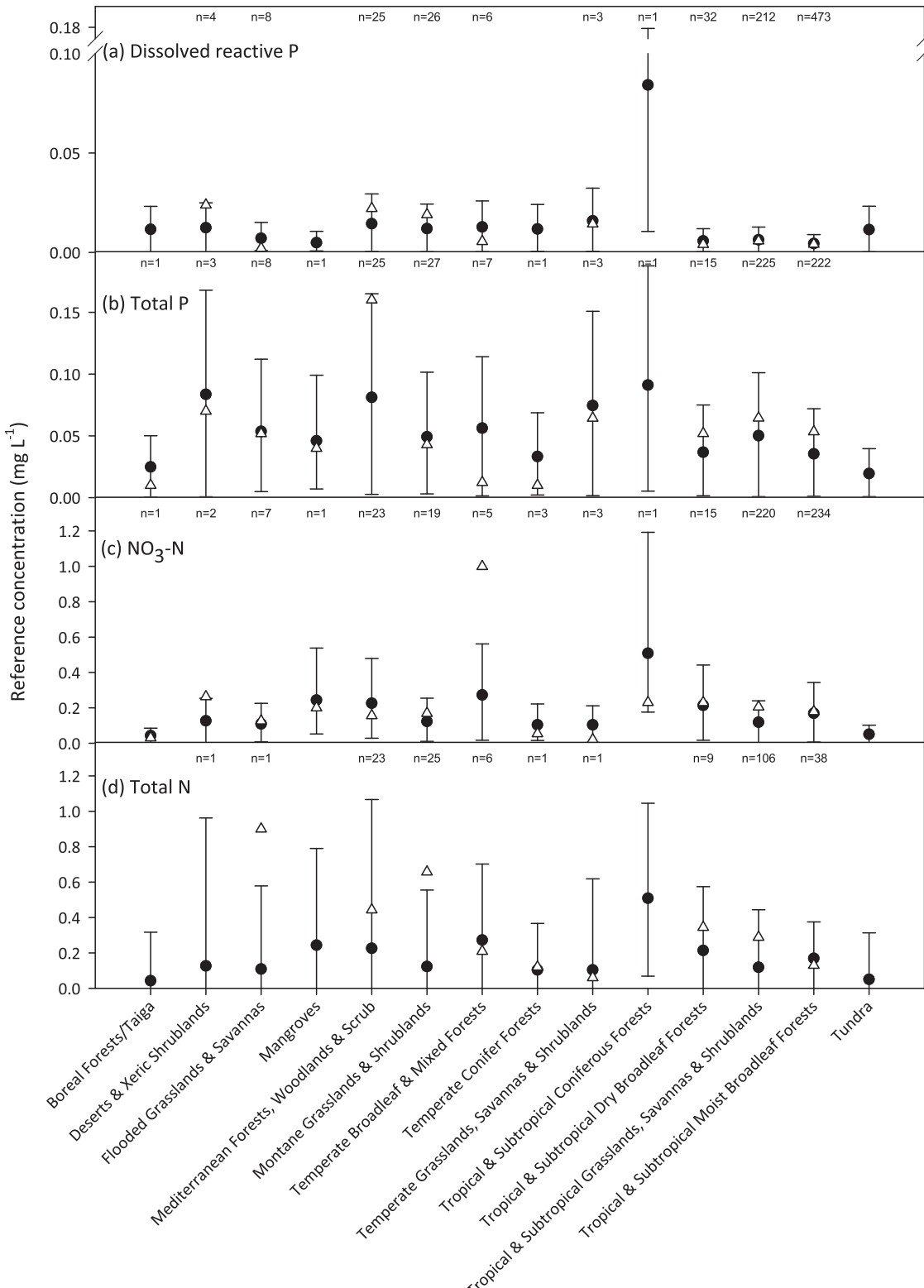

**Fig. 1 | Mean (± 95% confidence intervals) reference concentrations for each nutrient fraction by biome.** The fractions are (**a**) dissolved reactive phosphorus; **b** total phosphorus; **c** nitrate-nitrogen; and **d** total nitrogen. The triangles indicate the mean median concentration from the observational dataset for minimally disturbed catchments (set as Olsen $P = \leq 2\,\mathrm{mg\,kg^{-1}}$, population density of 0.001 persons $\mathrm{km^{-2}}$ and 0% cropland, pastureland or urban land). The labels beginning with "$n =$" refer to the number of catchments contributing to the mean of minimally disturbed catchment values.

(depending on nutrient form). Although there is always the potential that outputs were biased by the number of catchments in each biome, there was no difference between the proportion of observations across biomes and those present at level 6 of HydroBasins except for Deserts & Xeric Shrublands and Boreal Forests/Taiga biomes (see Supplementary Note 8). Owing to the paucity of observations in these two biomes, we elected not to include these data in continental statistics or maps.

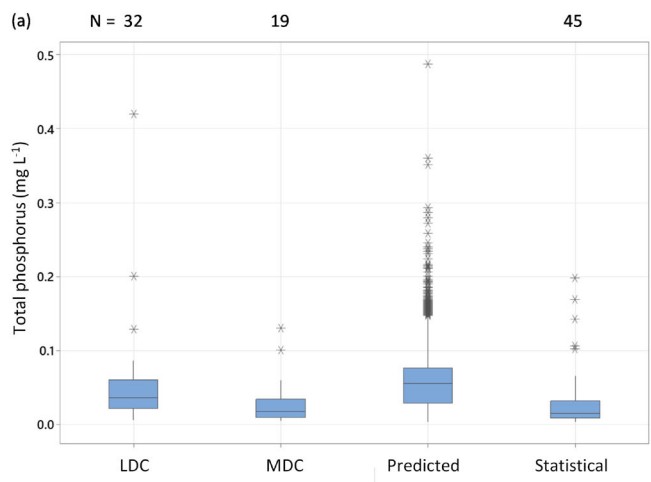

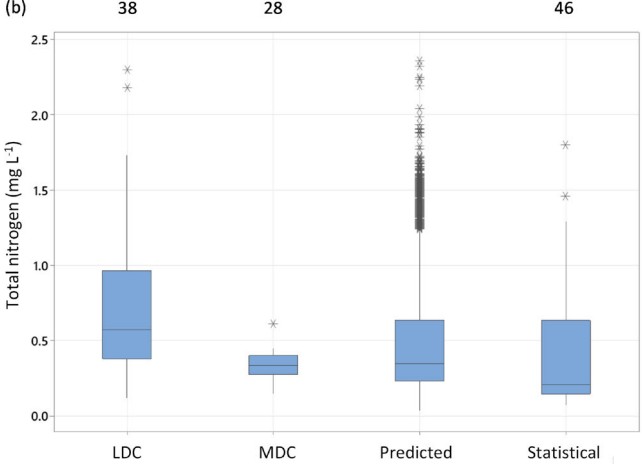

**Fig. 2 | Comparison of reference conditions by method of determination.**
The box plots show the count (*n*), and the 25, 50, 75th percentiles, the 5th and 95th percentiles as whiskers and outliers (as asterisks) of the concentration of (**a**) total nitrogen (TN) and (**b**) total phosphorus (TP) reference conditions in published (*n* = 37) literature (see Supplementary Table 9). Data are separated by the main method of determination (least disturbed condition [LDC], minimally disturbed condition [MDC], or statistical), and plotted against our predicted concentrations for level 6 of HydroBasins (*n* = 14241). A general linear model of the natural log-transformed concentrations of TN and TP indicated differences in reference

conditions by method for published TN but not for TP. A Tukey pair-wise comparison suggests that predicted concentrations are like those estimated by MDC or statistical methods but not LDC (for TN). Data for DRP and NO₃-N are included in Fig. S13. Least disturbed conditions represent a low (usually the 25th) percentile of concentrations for a representative set of sites, minimally disturbed conditions are sites confirmed to have little human influence (e.g., <5% agricultural land), and statistical refers to models that relate the level of human influence to nutrient concentrations and use this relationship to determine concentrations at zero human influence (i.e., the intercept of a regression relationship).

## Enrichment

We calculated enrichment as the difference between estimated current and estimated reference concentrations relative to reference conditions. The mean percentage of anthropogenic enrichment across nutrients varied from 2% for total phosphorus in Oceania to 123% for nitrate-nitrogen in Europe (Table 1 and Figs. 3–6). Europe had the highest percentage of enrichment across all analytes, followed by Asia and North America. The global average enrichment for nitrate-nitrogen and total nitrogen was 58 and 35%, and 21 and 14% for dissolved reactive phosphorus and total phosphorus, respectively. The mean percentage of nitrate-nitrogen as total nitrogen was 29 and 39% for reference and current conditions, while the same percentage for

dissolved reactive phosphorus as total phosphorus was 21 and 24%, respectively. Ref. 28 showed that an increasing population coupled with an expansion of agricultural land increases the percentage of dissolved inorganic nutrients relative to total nutrients which is consistent with our analysis.

Enrichment in Europe reflects a long history of high application of fertilisers (manure or inorganic), especially in Western Europe[29]. In Asia, enrichment is most pronounced in China, which has substantially increased inorganic fertilization over the past 40 years[30]. In North America, enrichment is greatest in the U.S. Midwestern states from intensive cropland. Interestingly, in both Asia and North America there appears to be a strong natural source of phosphorus (Figs. 4 and 6). This source correlates to the presence of naturally high-fertility soil types (e.g., Black Chernozems)[31], or frequent drought conditions. Drought can lower dilution rates but also increases soil hydrophobicity and infiltration-excess runoff. Drought followed by rainstorms can therefore lead to nitrogen- and phosphorus-rich sediment reaching rivers from increased surface runoff[32–34].

**Table 1 | Mean percentage (with 95% confidence intervals in parentheses) of continental and global enrichment derived from the proportion of the predicted median concentrations of each analyte (excluding polar regions and the Dessert & Xeric Biomes) minus the median reference values, relative to median reference concentrations[a]**

| Row labels | Nitrate-nitrogen | Total nitrogen | Dissolved reactive phosphorus | Total phosphorus |
|---|---|---|---|---|
| Africa | 50 (38–64) | 29 (21–42) | 25 (12–43) | 13 (5–27) |
| Asia | 82 (62–101) | 41 (30–56) | 24 (18–51) | 17 (8–33) |
| Europe | 123 (81–161) | 83 (60–105) | 48 (32–85) | 30 (15–47) |
| North America | 42 (26–59) | 30 (21–43) | 14 (8–32) | 10 (4–23) |
| Oceania | 6 (3–16) | 7 (3–17) | 5 (9–34) | 2 (1–11) |
| South America | 21 (14–33) | 14 (8–31) | 10 (6–34) | 9 (2–27) |
| World | 58 (41–75) | 35 (24–49) | 21 (14–46) | 14 (6–29) |

[a]Note that the percentage of catchments whose predicted reference value fell within the confidence intervals for current conditions was 0.36, 0.24, 0.22 and 0.39 for nitrate-nitrogen, total nitrogen, dissolved reactive phosphorus and total phosphorus, respectively. Owing to the use of a proportion, lower and upper confidence intervals were taken as the difference between the lower interval for predicted median minus the upper interval for the reference conditions, and the upper interval for predicted median minus the lower interval for the reference conditions, respectively.

## Potential for periphyton accrual

Predicted median nutrient concentrations were used in combination with the global mean total nitrogen and total phosphorus thresholds to categorize catchments across the globe as acceptable (predicted algal biomass <150 mg chlorophyll m⁻²) or unacceptable when greater than this amount based on prior research[35]. It should be noted that this analysis focuses on nutrient concentrations because they are directly affected by human activity. Although our analysis was restricted to baseflow in warmer months, we cannot fully account for variation in factors like temperature and light which may affect periphyton growth[36,37]. We then classified each catchment as 'acceptable periphyton biomass and nitrogen-limitation' (catchment type 1), 'undesirable periphyton biomass and nitrogen-limitation' (catchment type 2), 'acceptable periphyton biomass and co-limitation' (catchment type 3), 'undesirable periphyton biomass and co-limitation' (catchment type 4), 'acceptable periphyton biomass and phosphorus-limitation' (catchment type 5), or 'undesirable periphyton biomass and

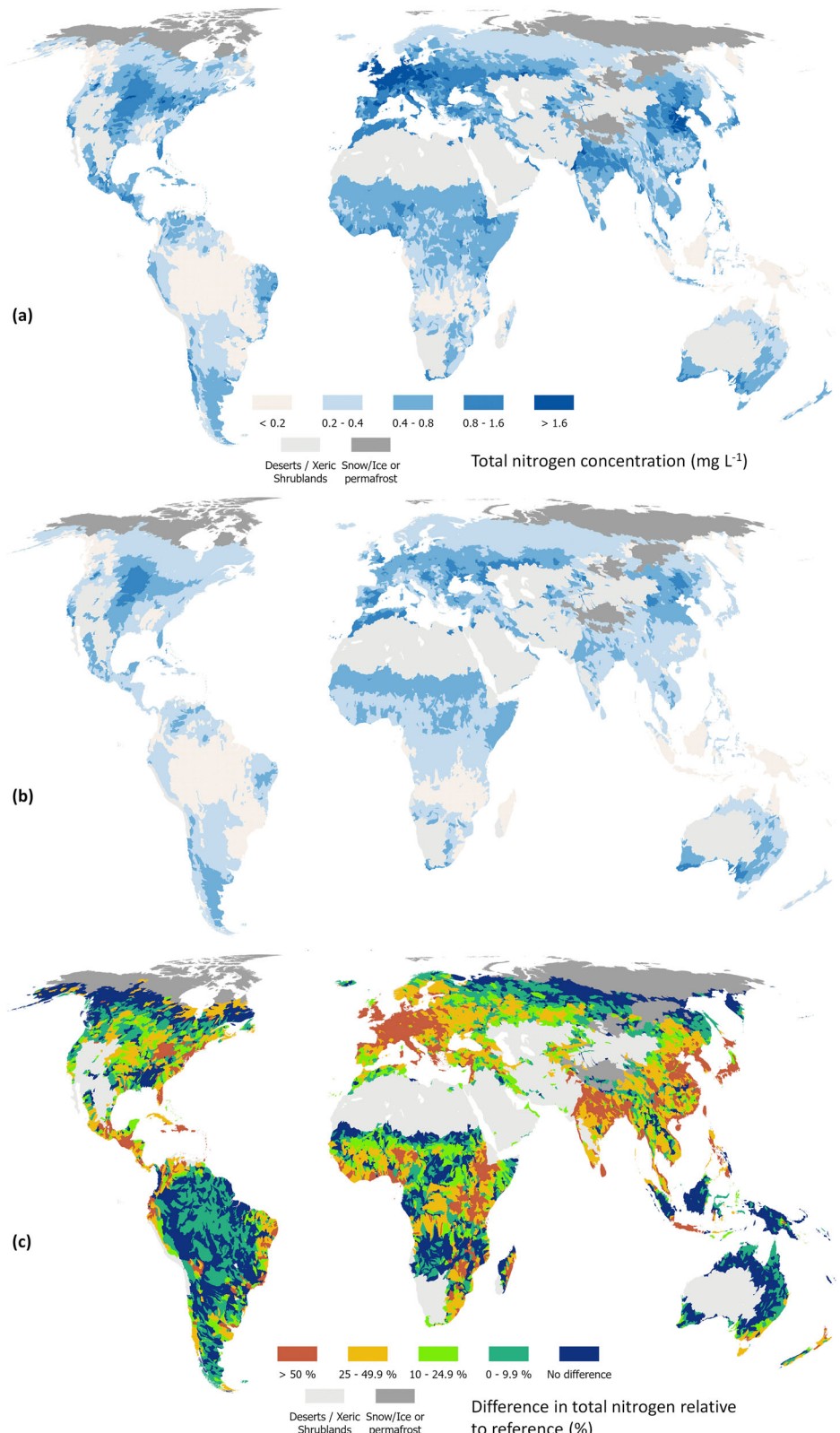

**Fig. 3 | Global distribution of total nitrogen (N) concentrations at level 6 of HydroBasins.** Concentrations are presented under current management (**a**), reference conditions (**b**) and as the difference between current and reference conditions expressed as a percentage of current conditions (**c**). Administrative boundaries from GADM using the Equal Earth projection (https://proj.org/en/stable/).

phosphorus-limitation' (catchment type 6). Under reference conditions <1% of the world had undesirable periphyton biomass limited by nitrogen or phosphorus and 2.7% by co-limitation, together these accounted for rivers where ~10% of the world's population (~777 million) live, relatively evenly spread across continents (Tables 2 and 3 and Fig. 7). However, under current conditions, ~1, 3 and 10% of the global landmass contained catchments likely to have undesirable levels of periphyton biomass caused by nitrogen, phosphorus or a

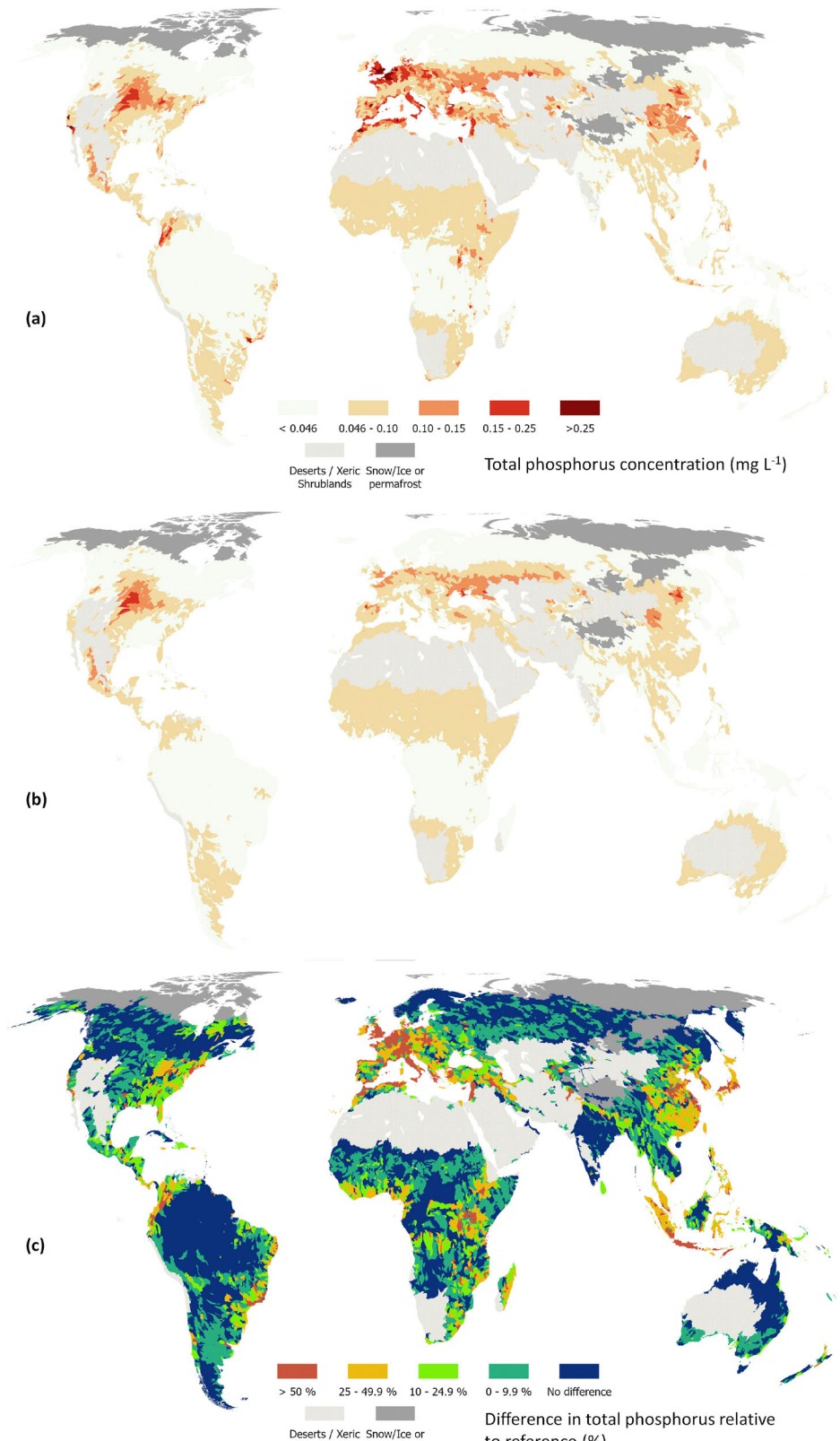

**Fig. 4 | Global distribution of total phosphorus (P) concentrations at level 6 of HydroBasins.** Concentrations are presented under current management (**a**), reference conditions (**b**) and as the difference between current and reference conditions expressed as a percentage of current conditions (**c**). Administrative boundaries from GADM using the Equal Earth projection (https://proj.org/en/stable/).

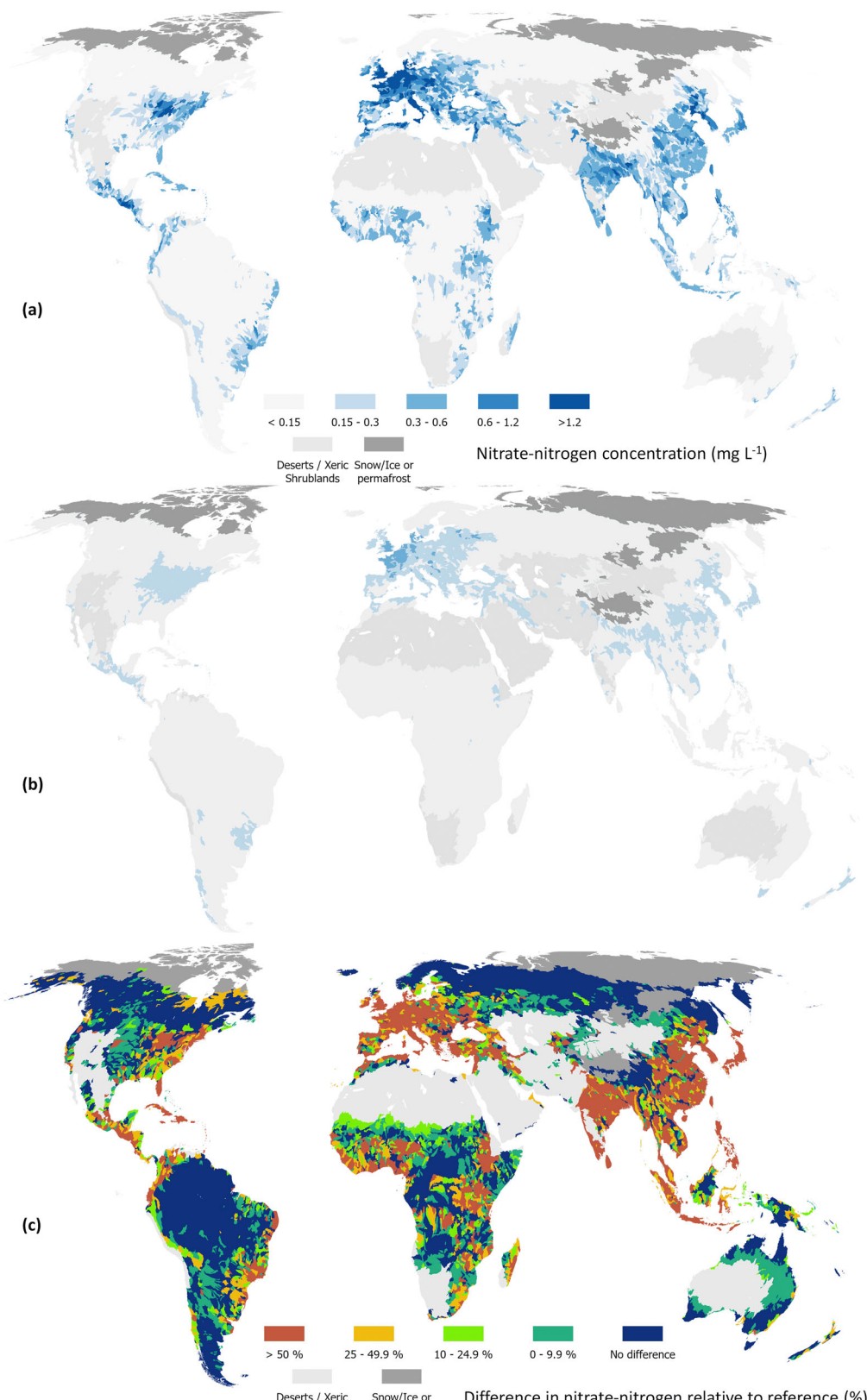

**Fig. 5 | Global distribution of nitrate-nitrogen concentrations at level 6 of HydroBasins.** Concentrations are presented under current management (**a**), reference conditions (**b**) and as the difference between current and reference conditions expressed as a percentage of current conditions (**c**). Administrative boundaries from GADM using the Equal Earth projection (https://proj.org/en/stable/).

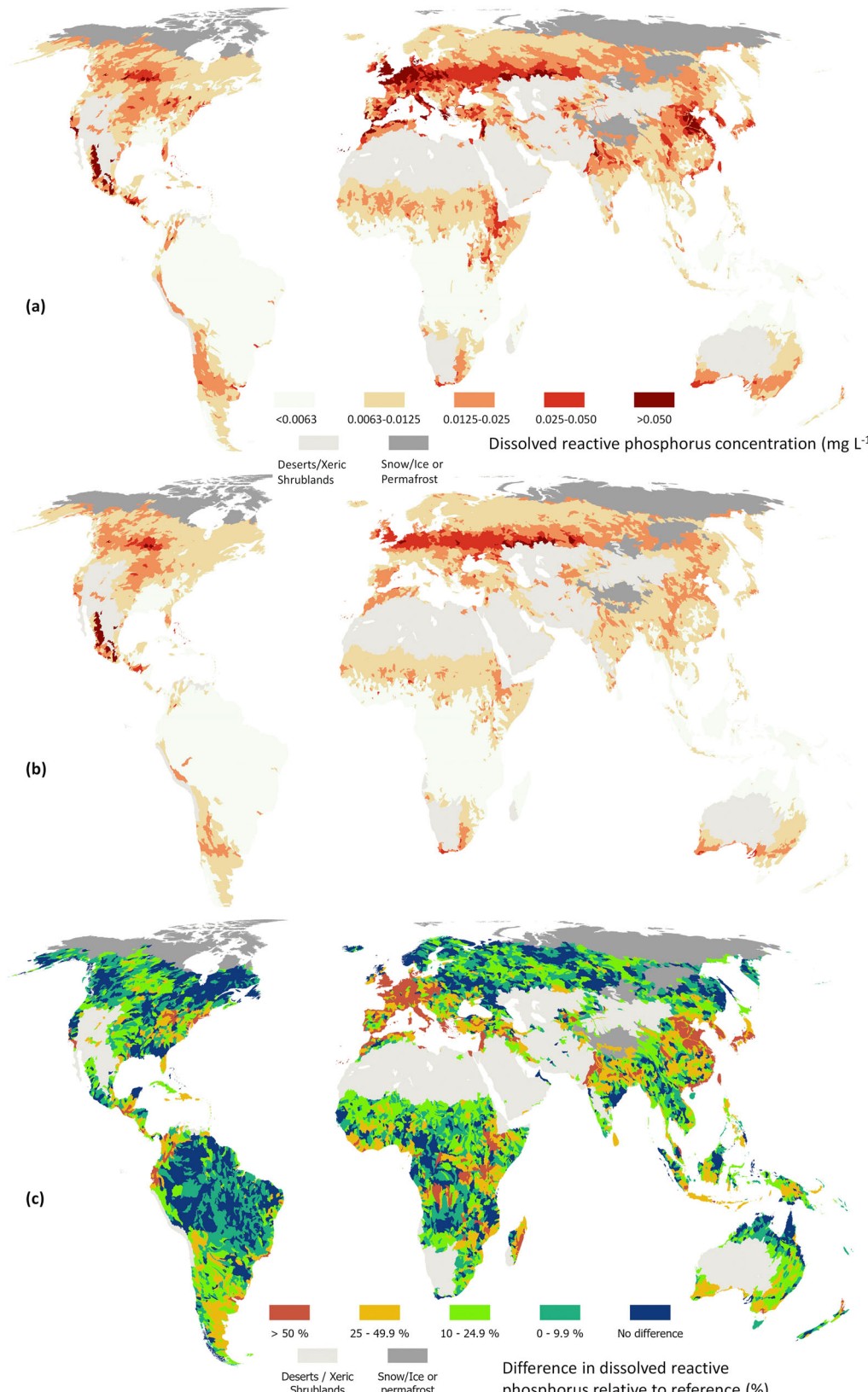

**Fig. 6 | Global distribution of dissolved reactive phosphorus (DRP) concentrations at level 6 of HydroBasins.** Concentrations are presented under current management (**a**), reference conditions (**b**) and as the difference between current and reference conditions expressed as a percentage of current conditions (**c**). Administrative boundaries from GADM using the Equal Earth projection (https://proj.org/en/stable/).

**Table 2 | Estimated percentage (with 95% confidence intervals in parentheses) of each continent and the world excluding polar regions and Dessert & Xeric Biomes predicted to have acceptable periphyton biomass and nitrogen-limitation (catchment type 1), undesirable periphyton biomass with nitrogen-limitation (catchment type 2), acceptable periphyton biomass and co-limitation (catchment type 3), undesirable periphyton biomass with co-limitation (catchment type 4), acceptable periphyton biomass and phosphorus-limitation (catchment type 5) and undesirable periphyton biomass with phosphorus-limitation (catchment type 6) under current and reference conditions**

| Continent | Condition | N-limited acceptable | N-limited undesirable | Co-limited acceptable | Co-limited undesirable | P-limited acceptable | P-limited undesirable |
|---|---|---|---|---|---|---|---|
| Africa | Current | 26.9 (15.7–37.8) | 0.8 (0.7–0.9) | 58.9 (66.8–51.1) | 4.5 (4.1–4.4) | 6 (8.4–3.7) | 2.2 (3.5–1.3) |
| | Reference | 60.1 (29.1–57.3) | 0.2 (0–0) | 34.7 (65.3–39.7) | 2 (1.4–2.1) | 1.5 (2–0) | 0.8 (1.5–0.2) |
| Asia | Current | 24.4 (18.6–30.9) | 1.3 (0.3–1.7) | 42.9 (42.4–43.1) | 10.9 (12.2–9.4) | 17.1 (21.2–12.9) | 2.8 (4.6–1.3) |
| | Reference | 41.5 (26.8–49.4) | 0.2 (0–0.8) | 43.3 (58.6–44.2) | 3.9 (4.3–2.2) | 10 (9.2–2.7) | 0.3 (0.5–0.1) |
| Europe | Current | 2.5 (1.2–4.7) | 2.9 (1–4.7) | 27.7 (22.5–34.3) | 33 (28.4–31.3) | 23 (28.3–17.5) | 10.9 (18.5–7.5) |
| | Reference | 37.3 (6.5–31.2) | 1.5 (0–2.1) | 46.6 (68.5–58.4) | 8.2 (14.5–4) | 6.4 (10.2–4.2) | 0 (0.2–0) |
| North America | Current | 11.5 (5.6–20.7) | 1.4 (0.2–2.7) | 54.2 (52.1–51.2) | 12.7 (13.3–9.8) | 18.9 (24.9–15.1) | 1.1 (3.7–0.2) |
| | Reference | 37.6 (9.9–37.6) | 1.4 (0.2–2.9) | 47.8 (67.2–48.8) | 1.1 (6.7–2.5) | 11.8 (15.1–7.9) | 0.1 (0.6–0) |
| Oceania | Current | 45.9 (37.8–53.4) | –[a] | 35.6 (36.2–35.1) | 2 (2.7–1.7) | 11.1 (15.9–5.6) | 4.6 (6.5–3.3) |
| | Reference | 37.2 (43.8–55.6) | – | 43.6 (35.3–39.2) | 2.4 (4.5–0.6) | 11.4 (12.7–0.2) | 4.6 (2.8–3.6) |
| South America | Current | 29.3 (19.7–38.1) | 0.1 (0–0.1) | 39.8 (41–43.2) | 0.6 (0.7–0.6) | 29.6 (37.5–18) | 0.5 (1.1–0) |
| | Reference | 36.1 (29.4–59.8) | – | 41.3 (55.1–39) | – | 22 (15.1–1.1) | 0.4 (0.2–0.1) |
| World | Current | 22.5 (15.2–30.5) | 1.1 (0.4–1.6) | 45.8 (46.8–44.9) | 9.7 (9.6–8.6) | 17.5 (22.5–12.3) | 2.9 (5.1–1.6) |
| | Reference | 43.4 (23.8–49.6) | 0.5 (0–0.9) | 42.1 (60.7–44.2) | 2.7 (4.3–1.9) | 10.3 (9.9–2.7) | 0.6 (0.8–0.3) |

[a]<0.01 million km² or 0.01% of the continental area.

**Table 3 | Estimated percentage (with 95% confidence intervals in parentheses) of the world's population (by continent and globally; note that currently the 2023 estimate of global population is 8.1 billion) predicted to have acceptable periphyton biomass and nitrogen-limitation (catchment type 1), undesirable periphyton biomass with nitrogen-limitation (catchment type 2), acceptable periphyton biomass and co-limitation (catchment type 3), undesirable periphyton biomass with co-limitation (catchment type 4), acceptable periphyton biomass and phosphorus-limitation (catchment type 5) and undesirable periphyton biomass with phosphorus-limitation (catchment type 6) under current and reference conditions**

| Continent | Condition | N-limited acceptable | N-limited undesirable | Co-limited acceptable | Co-limited undesirable | P-limited acceptable | P-limited undesirable |
|---|---|---|---|---|---|---|---|
| Africa | Current | 27.1 (12.5–37.5) | 7.8 (4.1–7.7) | 38.6 (48.7–32.6) | 14.3 (15–14.7) | 6.8 (9.4–3.7) | 5.5 (10.3–3.8) |
| | Reference | 60.3 (32.1–62.8) | 4.4 (0–0) | 28.6 (58.7–30.5) | 4.9 (6.1–6.7) | 1 (1–0) | 0.8 (2–0) |
| Asia | Current | 21.9 (14.8–27.6) | 1.8 (0.4–2.7) | 21.3 (23.8–21.9) | 21.3 (19.9–21.5) | 22.7 (27.6–19.6) | 11 (13.5–6.6) |
| | Reference | 47.5 (30.4–58.2) | 0.4 (0–0.2) | 29.4 (60.3–40.2) | 5.9 (3–1.4) | 15.1 (5.8–0.1) | 1.7 (0.6–0) |
| Europe | Current | 2.4 (1.3–5.2) | 6.4 (2.4–13.3) | 11.5 (9.3–14) | 51.5 (38.2–46.7) | 6.4 (8.9–4.9) | 21.7 (40–16) |
| | Reference | 47.1 (9.3–46.4) | 8.4 (0–1.7) | 31.8 (68.4–48.1) | 11.9 (19.8–3.7) | 0.8 (2.2–0.2) | 0 (0.2–0) |
| North America | Current | 11.9 (5.5–25.8) | 5 (3.7–6.4) | 37.9 (39.5–30.8) | 34.5 (30.5–29.8) | 7.3 (10.4–5.9) | 3.3 (10.3–1.2) |
| | Reference | 62.8 (22–77.9) | 4.1 (0–0.2) | 26.7 (73.9–20.2) | 5.1 (1.3–1.5) | 1.2 (1.8–0.2) | 0.1 (1–0) |
| Oceania | Current | 16.5 (15.3–25.8) | –[a] | 44.8 (35–44.9) | 5.6 (2–24.6) | 10.8 (13.2–3.2) | 22.3 (34.4–1.5) |
| | Reference | 21.2 (22.6–28.3) | – | 61.1 (62.2–65.9) | 8.6 (5.5–4.9) | 1.3 (8.3–0) | 7.7 (1.4–0.9) |
| South America | Current | 54.6 (38–66.8) | 5.8 (0–1.5) | 26.5 (35.5–25) | 5.1 (11.4–3.3) | 6.8 (10.7–3.4) | 1.2 (4.3–0) |
| | Reference | 73.3 (59.3–81.5) | – | 21.1 (36.8–18.2) | – | 4.2 (3.3–0.1) | 0.9 (0.5–0.2) |
| World | Current | 21.2 (13.3–28.2) | 3.7 (1.4–5) | 24.2 (27.3–23.4) | 24.3 (21.9–23.4) | 16.2 (20.3–13.5) | 10.4 (15.7–6.6) |
| | Reference | 51.8 (28.9–60.2) | 2.2 (0–0.3) | 29.1 (60.9–37.1) | 6.2 (5.2–2.3) | 9.5 (4.2–0.1) | 1.2 (0.8–0) |

[a]< 0.01% of the continental population.

combination of both nutrients being enriched, respectively. Together these catchments contain ~38% of the world's population (~3.1 billion people), with the greatest numbers living in Europe (80%) and North America (43%) (Table 3). Areas in an acceptable state under reference conditions that were limited by nitrogen, phosphorus or both nutrients amounted to ~52, 10 and 29% of the globe containing 43, 10 and 42% of the population, respectively. However, under current conditions, areas that were limited by nitrogen, phosphorus or both nutrients and had acceptable levels of periphyton biomass accounted for ~21, 16 and 24% of the global land mass, respectively (Table 2),

accounting for ~23, 18 and 46% of the world's population, respectively (Table 3).

The areas with co- or phosphorus-limitation and enrichment likely to induce undesirable periphyton biomass were predicted to occur in temperate regions with a history of phosphorus additions and intensive agriculture[38] (Fig. 4). However, there were also substantial areas in Africa and Australia that likely exhibit undesirable levels of biomass probably through a combination of low rainfall and high summer temperatures. Under reference conditions, the area with undesirable levels of periphyton biomass (largely caused by enrichment of both

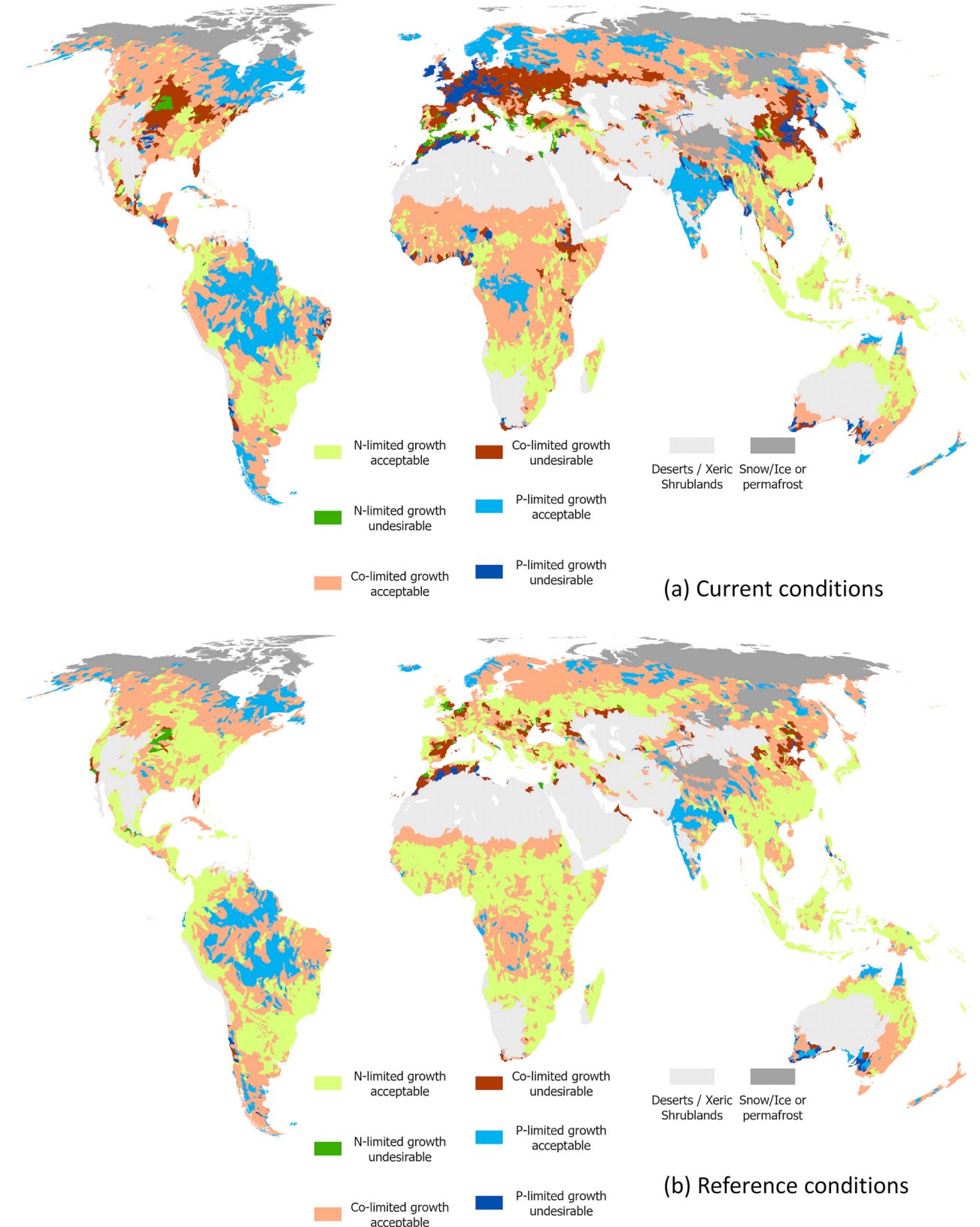

(a) Current conditions

(b) Reference conditions

nitrogen and phosphorus) decreased substantially, especially in Europe, North America, and Asia. Areas in Australia and north Africa were still likely to have undesirable levels of biomass owing to high temperatures and low rainfall.

Although many periphyton thresholds rely on total nutrients, as well as the equations we used to estimate periphyton accrual,

dissolved nutrients may be more available to periphyton[39]. This suggests that our predictions of periphyton accrual based on total nitrogen and phosphorus in rivers could underestimate the level of undesirable periphyton as population and resource use rates increase globally and shift N and P to more available forms. Additionally, while we focus here on attached algae, the nutrient enrichments can also

**Fig. 7 | Global map of the likelihood of periphyton growth.** Likelihood is presented for growth under (**a**) current and (**b**) reference conditions for nitrogen (N)-phosphorus (P)- or co-limitation when below or above a threshold concentration in nitrogen or phosphorus representing undesirable growth. Administrative boundaries from GADM using the Equal Earth projection (https://proj.org/en/stable/). Conditions for N-limited growth acceptable refers to N:P ratio is <7 and the median total N concentration is <0.800 mg L$^{-1}$. Conditions for N-limited growth undesirable refers to N:P < 7 and the median total N concentration is ≥ 0.800 mg L$^{-1}$. Conditions for Co-limited growth acceptable refers to N:P ≥ 7 and <15 and either the median total N concentration is <0.800 mg L$^{-1}$ or median total P concentration is <0.046 mg L$^{-1}$. Conditions for Co-limited growth undesirable refers to N:P ratio is ≥7 and <15 and both the median total N concentration is ≥0.800 mg L$^{-1}$ and median total P concentration is >0.046 mg L$^{-1}$. Conditions for P-limited growth acceptable refers to N:P ratio is ≥15 and the median total P concentration is <0.046 mg L$^{-1}$. Finally, conditions for P-limited growth undesirable refers to N:P ratio is ≥15 and the median total P concentration is ≥0.046 mg L$^{-1}$.

stimulate toxic suspended algal blooms in larger slower flowing rivers[40]. Our analysis likely underestimates the problems associated with nutrient enrichment in streams and future research could link models of suspended algal blooms in rivers and as influenced by nutrients with our approach[41].

## Discussion

Excessive algal biomass in streams can influence water quality, particularly given evidence that attached freshwater cyanobacteria in rivers can produce toxins and taste and odour problems that harm drinking water safety and quality[42]. These problems are arising worldwide[2,43]. Toxins can be transported from rivers to terrestrial or marine organisms as well[44]. Meaningful and attainable targets are therefore necessary to minimise the likelihood of unacceptable levels of biomass in rivers (and potentially to receiving lakes or impoundments). Assessing the current state of freshwater quality relative to reference conditions in large catchments is particularly useful in areas where local data is limited, e.g., to just cropland area. Our data can also be used as a screening tool to gauge the need for more locally relevant data and action. For example, more data could be used in two scenarios.

One scenario would see a regulator assessing the anthropogenic contribution that can be managed (by difference between current and reference concentrations) and the number and type of mitigations required to reduce periphyton to acceptable levels. Where mitigation is unable to reduce losses sufficiently, a decision may be necessary to reduce levels further by land use change or to accept that the catchment is used for agricultural production and will always have a level of unacceptable biomass[45].

A second scenario sees a regulator assessing current conditions and anthropogenic inputs and using these data to help to avoid setting targets that are close to reference conditions. At HydroBasins level 6, 24 and 39% of the catchments were within the confidence intervals of their predicted reference conditions for total nitrogen and total phosphorus, respectively, meaning that these catchments have low manageable losses and possibly naturally high periphyton biomass. These data may help regulators and landowners from spending large amounts of money on management or land use change that may not achieve desirable levels of periphyton biomass but does not excuse some spending and action to achieve a lower level of periphyton biomass that is above that likely under reference conditions.

In addition to informing policy, the manageable fractional nutrient increase can aid companies to isolate areas of their supply chain that cause environmental harm. Pressure from consumers and regulators combined with pressures for good corporate citizenship can lead multinational corporations to ask suppliers (e.g., farmers or processors) to lower their footprint. This is happening for greenhouse gas emissions and is projected to expand to other domains like freshwater quality (as measured by nitrogen and phosphorus losses)[46]. This pressure could lead to companies providing incentives to suppliers to implement mitigations (or land use change) to reduce nitrogen and phosphorus losses or penalties or (including market access restrictions) for high loss suppliers. By tackling water quality through both regulatory measures and incentives that respond to consumer demand, we can more quickly halt the decline of global freshwater quality associated with periphyton enrichment[47].

## Methods
### Dataset

Observations for concentrations of dissolved reactive phosphorus ($n$ = 305,741), total phosphorus ($n$ = 300,840), nitrate-nitrogen ($n$ = 297,783) and total nitrogen ($n$ = 308,118) checked, filtered, and harmonised to 2005–2013 from readily available, global (e.g., United Nations Environment Programme) or country-specific (e.g., United States Geological Survey or Land, Air, Water, Aotearoa) databases[48]. We chose 2005–2013 as a period of consistent data quality, to average across variation associated with continental or global climatic trends[49], and to minimize changing influences from factors such as changes in land use and land use practices[50]. The processes and justification used to check, filter, and harmonise the observations are available elsewhere[48]. Additional work[22] also matched and filtered sites in a catchment to those where concentrations were measured no more than 50 km upstream of a site where discharge was measured daily[51]. Discharge data were separated into baseflow and stormflow using a recursive digital filter[52] and used to filter out those samples taken where stormflow was ≥25% of mean daily discharge. We used this percentile to avoid including data taken during stormflow where periphyton was likely to be swept away or growth was likely inhibited by suspended sediment and light limitation[36].

We then filtered and generated medians (and the number of observations) of data for temperate and polar regions to values measured from May to October in the Northern Hemisphere and for November to April in the Southern Hemisphere. These corresponded to periods of likely peak growth[53]. Year-round data were used to generate medians and the number of observations for tropical regions. After checking, filtering, harmonising, and matching to baseflow conditions, the number of observations and catchments for dissolved reactive phosphorus were 78,896 and 2202, 108,188 and 2618 for total phosphorus, 87,662 and 2372 for nitrate-nitrogen and 38,092 and 1186 for total nitrogen, respectively.

Predictor variables relating to categorical and continuous factors known to affect nitrogen and phosphorus losses to water were sourced from a wide variety of sources and are listed in the Supplementary Table 1. These included several variables indicative of human activity such as land use and land use intensity[8,9,54,55]. Continuous variables like percentage cropland were expressed as an average across each catchment. However, where more than one categorical variable was present in a catchment (e.g., biomes), we used the category that covered the greatest area within the catchment.

### Modelling anthropogenic effects

The anthropogenic effect was estimated based on data for eight variables: population density, soil Olsen phosphorus concentration, and six land classes as given in Supplementary Table 1 and Supplementary Fig. 1. The anthropogenic effect is computed by the weighted sum of the measurements on each of these variables. The weights were modelled as coming from independent normal distributions with a common mean and variance to be computed as part of the model fitting process. The variables were normalised to have approximately similar variances and increase with greater anthropogenic activity. The six land class variables are all percentages, and their relative sizes correspond to their expected relative weightings. Land classes which

cover a large proportion of the area are expected to have more weight than land classes which cover only a small proportion of the area. However, to account for skewed distributions, percentage land classes were arcsine-square root-transformed, while also adjusting an increase in land uses associated with in anthropogenic activity such as land under crops, pasture or urban, with decreases in rangeland, forests, and lakes. For example, an increase in cropland, pastureland or urban land was expressed in rangeland as equal to the arcsine transformed ratio of (100-rangeland)/100). Population density and Olsen phosphorus have very skewed distributions so were log transformed. To make them similar in scale to the other variables, these variables were normalised to Olsen phosphorus = log(1.8578 × *Olsen phosphorus*) and Population density = log(0.0282 × *Population Density*). Following adjustments and normalisation, the standard deviations of Olsen phosphorus and population density were within the standard deviations of the other variables. In respect to various responses (e.g., total phosphorus, total nitrogen etc.), the eight variables relating to anthropogenic effect are further summarized by their first common latent component named 'human effect' using partial least squares[56]. We used this method to avoid potential effects from multi-collinearity amongst variables (especially land uses). The resulting distribution of human effect was normally distributed across each analyte. Human effect was calculated for each analyte as Eqs. (1–4) with correlation coefficients of 0.61, 0.48, 0.57 and 0.53, respectively:

$$
\begin{aligned}
&\text{Dissolved reactive phosphorus human effect} = 0.6482 \\
&\times \textit{Olsen phosphorus} + 0.3410 \times \textit{Pasture} + 0.1977 \times \textit{Lentic} + 0.4610 \\
&\times \textit{Population Density} + 0.3031 \times \textit{Forest} + 0.3113 \times \textit{Urban} - 0.1323 \\
&\times \textit{Rangeland} + 0.0759 \times \textit{Crop}
\end{aligned}
\tag{1}
$$

$$
\begin{aligned}
&\text{Nitrate} - \text{nitrogen human effect} = 0.2636 \times \textit{Olsen phosphorus} - 0.1019 \\
&\times \textit{Pasture} + 0.1031 \times \textit{Lentic} + 0.2264 \times \textit{Urban} + 0.4273 \times \textit{Crop} + 0.1119 \\
&\times \textit{Rangeland} + 0.3126 \times \textit{Forest} + 0.7520 \times \textit{Population Density}
\end{aligned}
\tag{2}
$$

$$
\begin{aligned}
&\text{Total phosphorus human effect} = 0.2468 \times \textit{Olsen phosphorus} + 0.0537 \times \textit{Pasture} \\
&+ 0.6571 \times \textit{Population Density} + 0.1068 \times \textit{Lentic} \\
&+ 0.5415 \times \textit{Urban} + 0.4055 \times \textit{Forest} + 0.0720 \\
&\times \textit{Rangeland} + 0.1736 \times \textit{Crop}
\end{aligned}
\tag{3}
$$

$$
\begin{aligned}
&\text{Total nitrogen human effect} = 0.6960 \times \textit{Olsen phosphorus} + 0.2229 \\
&\times \textit{Pasture} + 0.3031 \times \textit{Rangeland} + 0.0401 \\
&\times \textit{Crop} + 0.1548 \times \textit{Forest} + 0.3505 \times \textit{Urban} \\
&+ 0.1513 \times \textit{Lentic} + 0.4503 \times \textit{Population Density}
\end{aligned}
\tag{4}
$$

### Predicting water quality

We natural log-transformed concentration data and split it into a training and testing set, containing 75 and 25% of the total sites, respectively before modelling. The split ensures the distribution of the associated concentrations were similar between training, testing, and the original data (see Supplementary Table 2).

We used a random forest model to estimate the median of natural log transformed dissolved reactive phosphorus, total phosphorus, nitrate-nitrogen, and total nitrogen concentrations respectively for each site as a function of anthropogenic effects and variables capturing catchment characteristics (see Supplementary Table 1). We also included the number of observations used to derive a site median as the sample weight. A random forest model is an amalgam of classification and regression trees that partitions and minimises differences in water quality concentrations based on binary splits of predictor variables. The approach does not assume any distribution and is able to fit non-linear relationships and cope with the effects of high order interactions. The splits are made using a random subset of predictor variables to compare a bootstrapped sample of water quality concentrations. Including the random subsets increases the accuracy of

prediction when averaged over the forest. The models also maximise their accuracy of prediction for unseen data as the number of trees increases, meaning that the final model is less likely to be overfitted (here we use a tenfold change for cross-validation).

To find the best random forest model for each analyte concentration, we utilized the performance measurement mean absolute error (MAE) in two steps (see Supplementary Tables 3–7 and Supplementary Figs. 3–6). The first step used the training data to tune hyperparameters outputting the number of predictor variables available at each split with up to 900 trees per forest (as little improvement is found going beyond this number[57]) and a minimal node size. We started with the full model which included all the terms in the model, then searched for the best hyperparameters based on the parameter grid by minimum MAE and a tenfold change cross-validation. To tune hyperparameters, we dropped the lowest importance predictor to get a second model and repeated this process until only the fixed parameters remained. As our aim was to determine the effect of humans on water quality concentrations and to extend the reference values (at near zero anthropogenic effect) to biophysically similar regions we fixed Latitude, Human effect, and Biome into the models. For each concentration we obtained 23 models with tuned hyperparameters.

In the second step we applied each of the 23 models to the testing data and calculated the MAE. We choose the model with smallest MAE and where models were similar, the fewest terms. The array and performance of models assessed for each nutrient is given in see Supplementary Tables 3–7.

Final model performance was assessed by comparing observations with predicted values for the training and (independent) testing datasets. Terms in the final model along with their relative importance are given in Supplementary Figs. 3–6. The performance of the final models was summarised as the coefficient of determination ($R^2$) of a regression of observations vs predictions, and the root mean square error which represents the mean deviation between predicted and observed values (Supplementary Table 7). Confidence intervals (95%) for each model with and without human effects were estimated by bootstrap resampling one thousand times, with replacement.

### Spatial implementation of modelling outputs

The final validated models were used to predict median concentrations of dissolved reactive phosphorus, nitrate-nitrogen, total phosphorus, and total nitrogen globally in ArcGIS (v10.1, Redlands, CA) under current conditions and under reference conditions. To estimate reference conditions, the terms within equations for Human Effects were set at Olsen phosphorus = 2 mg kg$^{-1}$ (effectively unfertilised soil under native land use[58]), a population density of 0.001 persons km$^{-2}$, and 0% cropland, pasture land or urban land (with an equal split between forest or rangeland).

The final validated models were converted from the R-programming language into Python for implementation within ArcGIS to extend predictions out to areas without observations. The performance of the Python implementation of the tuned models was similar but not identical to the development in R because different training and test data are inevitably chosen. The outputs are given in Supplementary Table 8 and Supplementary Figs. 7–12.

In addition to checking model performance, we also checked each model's spatial predictions by plotting the residuals (see Supplementary Fig. 13) and predictions of reference conditions by comparing predictions to those catchments under MDC (i.e., most likely to be under reference conditions in our database). We defined MDC as: Olsen phosphorus concentrations of ≤5 mg kg$^{-1}$, population density of ≤0.001 persons km$^{-2}$ and ≥80% forestland or rangeland and no cropland or urban land. Note that this is different from 'reference conditions' defined above to allow more sites to qualify as under MDC. The

model was considered to have adequately predicted reference conditions if those catchments under reference conditions fell within the 95% confidence interval for its respective biome.

From the Python implementation, raster grids were created with a spatial resolution of 0.025°, which corresponded to the coarsest grid cell associated with the input data listed in Supplementary Table 1. These grids were then used with the input variables in Supplementary Table 1 and the validated models to predict concentrations for sites at HydroBASINS level 6 ($n = 14241$ catchments[59]). This level was equivalent to the median level of our catchments.

We used these predictions to generate two outputs. The first was to estimate the percentage anthropogenic effect at each site taken as the ratio of the difference between predicted current and reference concentrations and predicted current concentrations. Although the raw data are given for each site and catchment, we also summarised these data as the area-weighted mean in each biome and included estimates of variation and confidence intervals so that policy makers with a paucity of data could determine the likelihood of anthropogenic effects and the need to remediate them.

The second output was to extend published work[4] that looked at the likelihood of nuisance levels of periphyton biomass in streams and rivers under reference conditions based on a Redfield nitrogen:phosphorus ratio of 7:1 (by mass) and the mean thresholds of periphyton biomass for total nitrogen and total phosphorus in the water column. Owing to variation in the response of periphyton to either nitrogen or phosphorus limitation we adjusted ratio to <7 and ≥15 for nitrogen and phosphorus limitation, respectively, and introduced classes for co-limitation by both nutrients whose ratio was ≥7 and <15[36,60–62]. We note that at high nutrient concentrations the relationship to periphyton biomass is weaker, but nonetheless still significant[53]. We used the equation of ref. [53] derived from relationships between total nitrogen and phosphorus (from the United States, New Zealand, Australia, Canada, and Europe) and maximum chlorophyl-a biomass ($\log(max.chl) = 0.349.\log(total\ nitrogen) + 0.256.\log(total\ phosphorus) + 0.722$) to confirm that our concentrations of total nitrogen (0.800 mg L$^{-1}$) and phosphorus (0.046 mg L$^{-1}$) met expectations in the literature for the boundary between acceptable and undesirable biomass (-150 mg chl-a m$^{-2}$)[35,63,64].

We classified each catchment into one of six classes. Rules for class one were if the nitrogen:phosphorus (as total nitrogen and total phosphorus) ratio is <7 and the median total nitrogen concentration is <0.800 mg L$^{-1}$, the site will have a quantity of periphyton biomass that is acceptable and is nitrogen-limited (catchment type 1). Rules for class two were if the nitrogen:phosphorus (as total nitrogen and total phosphorus) ratio is <7 and the median total nitrogen concentration is ≥0.800 mg L$^{-1}$, the site will have a quantity of periphyton biomass that is undesirable and is nitrogen-limited (catchment type 2). Rules for class three were if the nitrogen:phosphorus (as total nitrogen and total phosphorus) ratio is ≥7 and <15 and either the median total nitrogen concentration is <0.800 mg L$^{-1}$ or median total phosphorus concentration is <0.046 mg L$^{-1}$ the site will have a quantity of periphyton biomass that is acceptable and is co-limited (catchment type 3). Rules for class four were if the nitrogen:phosphorus (as total nitrogen and total phosphorus) ratio is ≥7 and <15 and both the median total nitrogen concentration is ≥0.800 mg L$^{-1}$ and median total phosphorus concentration is > 0.046 mg L$^{-1}$ the site will have a quantity of periphyton biomass that is acceptable and is co-limited (catchment type 4). Rules for class five were if the nitrogen:phosphorus (as total nitrogen and total phosphorus) ratio is ≥15 and the median total phosphorus concentration is <0.046 mg L$^{-1}$, the site will have a quantity of periphyton biomass that is acceptable and is phosphorus-limited (catchment type 5). Finally, rules for class six were if the nitrogen:phosphorus (as total

nitrogen and total phosphorus) ratio is ≥15 and the median total phosphorus concentration is ≥0.046 mg L$^{-1}$, the site will have a quantity of periphyton biomass that is undesirable and is phosphorus-limited (catchment type 6).

We removed catchments in biomes where rivers were less likely to be permanently flowing due to a lack of rainfall (e.g., Desert & Xeric shrublands) or have originated from areas with a 50–90 or >90% chance of being under permafrost[65] (e.g., Taiga).

Outputs for each nutrient concentration under current and reference conditions, along with the class for each catchment is given in an interactive map (https://global-water-quality.agr.nz/).

## Data availability
The unprocessed data used in this study are available from the United Nations Environment Programme (https://gemstat.org/), United States Geological Survey (https://www.waterqualitydata.us/), Land, Air, Water Aotearoa (https://www.lawa.org.nz/), and Global Runoff Data Centre (https://grdc.bafg.de). All processed data used in our models and that support the main findings of this study have been deposited in Figshare (https://doi.org/10.6084/m9.figshare.24188787.v1).

## Code availability
The Python code used in the filtering of GIS data and the resulting geotiff files presented in this paper is available in Figshare (https://doi.org/10.6084/m9.figshare.24188787.v1).

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

## Acknowledgements

The authors acknowledge the various organisations that provided data for this study (e.g., United Nations Environment Programme, United States Geological Survey, Land, Air, Water Aotearoa, Global Runoff Data Centre).

## Author contributions

R.W.M. obtained the data, conceived and wrote the paper and helped conduct the analysis. D.L. and M.U. curated the data and helped conduct the analysis. P.P. did the geospatial analysis. W.K.D. co-wrote the paper.

## Competing interests

The authors declare no competing interests.
