## [Transparent Peer Review file · Nature Communications]

Anthropogenic nutrient inputs cause excessive algal growth for nearly half the world's population

Corresponding Author: Professor Rich McDowell

Version 0:

Reviewer comments:

Reviewer #1

(Remarks to the Author)

This manuscript uses a comprehensive global dataset of river nutrient concentrations and associated land use / population data to predict the likely river reference conditions across the world. It then compares these with the current conditions, to assess the impact of anthropogenic activities on potential algal growth. It is a really original idea, and an interesting and well-written paper, and offers a way to determine human pressure in regions with poor data coverage. I think that the authors need to address some issues before publication.

Major comments

1) The assumption that high nutrients result in high periphyton accrual (line 72) is probably true for lakes, but the case for rivers is much weaker. What evidence do the authors have that nutrient concentrations are directly related to periphyton biomass or chlorophyll concentration? I would assume that flow, light, shading levels, water temperature and river length also have a major role in determining algal biomass.

2) The authors have used the Redfield Ratio to determine if algae are P or N co-limited. This may work in relatively pristine environments with little nutrient enrichment, but is quite meaningless in regions with high anthropogenic pressures. In highly-enriched river environments, neither P or N will be limiting algal growth, and they are likely to be light or temperature limited. The authors should have an additional category of "no nutrient limitation".

Minor comments

Title: Change to Anthropogenic nutrient inputs to rivers cause.....

Line 47: Should MDC also include a human population threshold? Wastewater inputs are more important than agricultural in many regions.

Line 95: Could P concentrations related to slope also relate to cities being mainly being situated on flat land?

Line 150: The statement that algal biomass in rivers is directly related to nutrient concentration is a major assumption that needs justifying (or caveats need to be added).

152: The periphyton accrual threshold is given as 150 mg/L. I can understand how phytoplankton can be quantified in mg chlorophyll /L, but should attached algae be mg/m²?

185: The paper focusses on potential for periphyton blooms, but phytoplankton blooms are surely the key parameter to measure, in terms of ecology and human health?

Materials and methods, Line 7: Please give the reader some indication of where these data sources generally come from, rather than asking them to follow up the citation.

Reviewer #2

(Remarks to the Author)

Overview

The authors have carried out an analysis of nutrient data to determine a global assessment of the extent of nutrient enrichment of rivers. The analysis uses a selection of environmental variables that provide both an assessment of anthropogenic nutrient input and factors that are likely to influence transport to the riverine system. They determine enrichment using a ratio of predicted values to reference values derived from the same models in catchments subject to minimal human influence.

The modelling approach is comprehensive and well executed. Anthropogenic influence was determined using a derived variable following partial least squares regression. The output from this modelling step was fully described in the Supplementary Material and was consistent with both my experience and the published literature and thus likely to be an appropriate indicator for inclusion in the subsequent modelling. To predict nutrient concentrations Regression tree models were used to select key variables which were then used to make predictions. The modelling approach split data into training and independent data sets, an important step not always used. To extend the output to other geographic areas a GIS based modelling approach was used. This approach produced similar but not identical variable selection and as a consequence different predictions. Literature data were used to determine the likelihood of excess algal growth in conditions of N, P or co-NP limitation using threshold concentrations of TN and TP.

The key results presented in the main body of the paper were

1. Proportion of global enrichment split by region (Table 1)
2. Continental estimates of the area of catchments & % of world population likely to have unacceptable/acceptable growth of periphyton

These results are presented in both tabular and graphical form.

Validity

As a non-professional statistician my assessment is that the approach used was valid and well executed. The methods section and the supplementary material provide adequate information to support the conclusions. Having attempted analysis of large scale data sets I appreciate the difficulties involved in data cleaning. I was not able to assess this aspect of the approach as details were only available in a referenced publication, but I have no reason to believe that this will not have been well executed.

Significance

There is clearly a very substantial literature describing the impacts of nutrient enrichment on the water environment and thus it is not surprising that the paper concludes that anthropogenic enrichment affects nearly 50% of the world's population. In that sense there is nothing particularly original in this paper's findings. However, it does present an interesting and in my opinion analysis of a very large global data set.

The paper starts with a good, although not particularly novel, overview of the concept of reference conditions. This is important as the authors use deviation from reference conditions to assess impact. It is this approach that I feel makes the paper worthy of publication.

Improvements

Key Results

In my opinion the most important results in this paper were the predictions of reference conditions derived from the models. However, these were not the main focus of the paper and I suggest that this needs to be addressed in a significant modification to the paper

I feel that the paper needs to include more consideration of the reference condition predictions in the main body of the paper. Currently all of the detailed information is only within the supplementary material. My suggestion is that it is this aspect of the work that is novel and needs to be brought into the main body.

Figure S13 provides much useful information and would warrant further discussion. There is currently much uncertainty over the variation of reference conditions, perhaps supported but another of the authors findings in their analysis of published values (i.e. the significance of country and method), also the importance of latitude in the main models. I would like to see more of this in the main paper.

Summary

In overview I found the paper interesting and well executed. However, in its current form I don't feel it provides sufficient new information to warrant publication, although it has conducted a worthwhile analysis of a very substantial data set. If the paper were to focus more on the issue of reference conditions in the context of geographic variability and their importance in impact assessment I do feel the paper could be worthy of publication

Other comments

Line 84 Sub-title. I did not find this sub-title particularly helpful, it needs to be modified

Line 128 enrichment varied from 3% to 86% (is it not 2% for TP Table 1)

Line 338 I found the legend to table 1 difficult to understand and consequently was not entirely clear what the table contained. Are the values proportions of catchments or the median enrichment level (O-E)/E derived from the predicted reference values.

Reviewer #3

(Remarks to the Author)

I found this paper very impressive. The most noteworthy results for me were how low both nitrogen-nitrate and total phosphorus need to be to avoid unacceptable periphyton concentrations. I think particularly the low nitrate is indicative of the prevalence of N fixing periphyton, requiring much lower concentrations in nitrate. The low concentrations of phosphorus are likely unattainable for most urbanised catchments but this paper at least provides evidence for the requirements of lower P concentrations and how much work there is left to be done. To have modelled across the globe further increases the

noteworthy-ness of the work and the high impact.

The work is wide ranging because of the variety of fields that feed into the human impact. The combined influence of urban pressures and intensification of agriculture with the various subfactors was an interesting approach as often the human effect is segregated into rural versus non-rural pressures. It might have been useful since so much work has been done to make a stab at the impact of those sub categories to give more clarity of what pressures need to be addressed. I think fields that are related to the sub categories, for example Environmental Planners, might feel that they need more insight into the proportion that population density is impacting the nutrient enrichment, for example. As the developed world puts limits on housing development in sensitive areas, this work could provide real evidence to support those often unpopular decisions that have been taken in catchments across the UK for example.

The discussion section was relatively short. I would suggest this is the weakest part of the paper and I think some more thinking could be done about the implications of the work that has been completed. The feat of processing 1.2 M data and making a global assessment of nutrient enrichment is rather exciting. The potential for using these findings are wide ranging - some more of that impact would be great to see in the paper.

There were a couple of areas that I either just didn't get or think could be improved. In the list of describing the 6 types of catchments, in description 4 you say if "both" conditions are met but then you say "or" when listing the conditions. I would have thought this should be "and".

I really like how much care was taken with training datasets, testing sets, checking residuals, looking at patterns. Often this much care is missing from papers and leads to weakness in the modelling. Having read all the supplemental info and the modelling approach, I have confidence in your methods and the care you've taken. One thing I was wondering is whether it's possible to get an idea of how confident you are in your values for enrichment. You have stated values like the global-average enrichment value for nitrate was 21% but this really doesn't mean a lot to me without confidence statements. If your confidence range is something like 15-28% then that's quite different to 5-38%. Also in Table S9, which nutrient are you referring to in column 5? I'd also expect an comment on some clear outliers in the comparison between estimated and actual values (Fig S11). I appreciate they're both in the supplemental data but I haven't found mention of any particular outliers in the main body text - only mention of catchments that were removed due to a lack of water.

Overall though I found the description of methods and the process excellent - and if this paper is published then something I'd be able to teach my students. I think the main area to improve on is the significance of the results and how they impact the topics chosen for discussion. Perhaps widening the discussion a little to describe that impact will achieve this.

Version 1:

Reviewer comments:

Reviewer #1

(Remarks to the Author)

Reviewer #2

(Remarks to the Author)

This is now an excellent paper, in my opinion the modifications made meet all of my and other reviewers comments. Well done

Point-by-point response to the reviewers' comments

Reviewers' Comments:

Reviewer #1 (Remarks to the Author)

This manuscript uses a comprehensive global dataset of river nutrient concentrations and associated land use / population data to predict the likely river reference conditions across the world. It then compares these with the current conditions, to assess the impact of anthropogenic activities on potential algal growth. It is a really original idea, and an interesting and well-written paper, and offers a way to determine human pressure in regions with poor data coverage. I think that the authors need to address some issues before publication.

Major comments

- 1) The assumption that high nutrients result in high periphyton accrual (line 72) is probably true for lakes, but the case for rivers is much weaker. What evidence do the authors have that nutrient concentrations are directly related to periphyton biomass or chlorophyll concentration? I would assume that flow, light, shading levels, water temperature and river length also have a major role in determining algal biomass. **Good point, although we filtered the data for warmer months under baseflow, we did not account for variation in light or temperature. We have added a caveat to this effect in the text on "Potential for periphyton accrual". See Minor comment 4. While it is true that the relationship between planktonic chlorophyll in lakes is much stronger than it is for rivers and streams, there is a significant relationship, nonetheless. This has been established by several meta-analyses. Here are a few references to illustrate the point.**
 - Biggs, B. J. F. 2000. Eutrophication of streams and rivers: dissolved nutrient-chlorophyll. *Journal of the North American Benthological Society* **19**:17-31.
 - Busse, L. B., J. C. Simpson, and S. D. Cooper. 2006. Relationships among nutrients, algae, and land use in urbanized southern California streams. *Canadian Journal of Fisheries and Aquatic Sciences* **63**:2621-2638.
 - Dodds, W. K., V. H. Smith, and K. Lohman. 2006. Nitrogen and phosphorus relationships to benthic algal biomass in temperate streams *Canadian Journal of Fisheries and Aquatic Sciences* **63**:1190-1191.
 - Munn, M., J. Frey, and A. Tesoriero. 2010. The Influence of Nutrients and Physical Habitat in Regulating Algal Biomass in Agricultural Streams. *Environmental Management* **45**:603-615.
- 2) The authors have used the Redfield Ratio to determine if algae are P or N co-limited. This may work in relatively pristine environments with little nutrient enrichment, but is quite meaningless in regions with high anthropogenic pressures. In highly-enriched river environments, neither P or N will be limiting algal growth, and they are likely to be light or temperature limited. The authors should have an additional category of "no nutrient limitation". **We accept that temperature and light will likely play an important role in limiting periphyton production and have reiterated this as per our response to the previous points. However, it turns out that the relationship between growth and nutrient concentrations it is not meaningless, just a weaker effect at higher nutrient levels. Below is a**

figure from the meta-analysis of Dodds, et al 2006. Canadian Journal of Fisheries and Aquatic Sciences **63**:1190-1192. You can see that benthic chlorophyll keeps increasing even as total phosphorus reaches very high levels. It also makes it evident that there is a weaker relationship between water column nutrients and benthic chlorophyll as P gets very high, but there still is a relationship. Hence, we are not quite sure that we should create a new category of no nutrient limitation, but note the possibility in the text, as at least there is less nutrient limitation. Also note that this analysis makes it clear that N:P matters and at higher nutrient levels, the presence of the other nutrient stimulates biomass accrual.

Fig. 2. Relationships between total P and mean benthic chlorophyll *a* as a function of the total nitrogen to total phosphorus (TN:TP) mass ratio in stream water for (a) the literature data set and (b) the United States Geological Survey data set. Note that (b) is not reprinted here as it remains unchanged from the original publication. Lowess curve fitting was used to fit points in this plot to illustrate how chlorophyll yield varies with TN:TP. Open circles and dotted lines are for points with TN:TP less than 15 by mass, solid triangles and solid lines represent points with TN:TP greater than 15.

Minor comments

1. Title: Change to Anthropogenic nutrient inputs to rivers cause..... **Change made.**
2. Line 47: Should MDC also include a human population threshold? Wastewater inputs are more important than agricultural in many regions. **Yes, MDC should include low (i.e. <5%) urban land use and also minimal wastewater treatment discharges. We have corrected this. We also mention in the Methods that sites were removed sites if located within 50km of a wastewater treatment plant.**
3. Line 95: Could P concentrations related to slope also relate to cities being mainly being situated on flat land? **Yes, that is a possibility. We've now included it in the following sentence " Our analysis indicated a strong influence of intensively farmed land, enriched soil fertility, and**

greater population density (e.g., especially on flat land) on increased river nutrient concentrations, as have others”.

4. Line 150: The statement that algal biomass in rivers is directly related to nutrient concentration is a major assumption that needs justifying (or caveats need to be added). Correct, we have added the following two sentences and references just after the opening sentence “It should be noted that this analysis focuses on nutrient concentrations because they are directly affected by human activity. Although our analysis was restricted to baseflow in warmer months, we cannot fully account for variation in factors like hydrology, temperature, and light which may affect periphyton growth”. We note that problems with excessive algal biomass generally occur during baseflow and during warmer months. Francoeur, S. N., Biggs, B. J. F., Smith, R. A. & Lowe, R. L. Nutrient limitation of algal biomass accrual in streams: seasonal patterns and a comparison of methods. *J. N. Am. Benthol. Soc.* 18, 242-260 (1999). Hill, W. R. & Fanta, S. E. Phosphorus and light colimit periphyton growth at subsaturating irradiances. *Freshwat. Biol.* 53, 215-225 (2008).
5. 152: The periphyton accrual threshold is given as 150 mg/L. I can understand how phytoplankton can be quantified in mg chlorophyll /L, but should attached algae be mg/m²? Yes, and thanks, that was a mistake – corrected to m⁻².
6. 185: The paper focusses on potential for periphyton blooms, but phytoplankton blooms are surely the key parameter to measure, in terms of ecology and human health? Walter, your thought on this one? We agree this can be important in some cases. Predominantly very slowly flowing rivers (with low dilution rates) without high values of suspended inorganic sediments (which intercept light) are the places where phytoplankton blooms are problematic. There are some prominent rivers (e.g. Murray-Darling in Australia, the Ohio River in the United States) where this has caused serious problems. However, we only know of one publication that starts to make this link (Van Nieuwenhuysse, E. E. & Jones, J. R. Phosphorus chlorophyll relationship in temperate streams and its variation with stream catchment area. *Can. J. Fish. Aquat. Sci.* 53, 99-105 (1996)). We had addressed this at line 220 where we state “Additionally, while we focus here on attached algae, the nutrient enrichments can also stimulate toxic suspended algal blooms in larger slower flowing rivers⁴⁰. Our analysis likely underestimates the problems associated with nutrient enrichment in streams and future research could link models of suspended algal blooms in rivers and as influenced by nutrients with our approach⁴¹.”
7. Materials and methods, Line 7: Please give the reader some indication of where these data sources generally come from, rather than asking them to follow up the citation. Ok, examples of global and country-specific databases have been included.

Reviewer #2 (Remarks to the Author)

Overview

The authors have carried out an analysis of nutrient data to determine a global assessment of the extent of nutrient enrichment of rivers. The analysis uses a selection of environmental variables that provide both an assessment of anthropogenic nutrient input and factors that are likely to influence transport to the riverine system. They determine enrichment using a ratio of predicted values to reference values derived from the same models in catchments subject to minimal human influence. The modelling approach is comprehensive and well executed. Anthropogenic influence was determined using a derived variable following partial least squares regression. The output from this modelling step was fully described in the Supplementary Material and was consistent with both my experience and the published literature and thus likely to be an appropriate indicator for inclusion in the subsequent modelling. To predict nutrient concentrations Regression tree models were used to

select key variables which were then used to make predictions. The modelling approach split data into training and independent data sets, an important step not always used. To extend the output to other geographic areas a GIS based modelling approach was used. This approach produced similar but not identical variable selection and as a consequence different predictions. Literature data were used to determine the likelihood of excess algal growth in conditions of N, P or co-NP limitation using threshold concentrations of TN and TP. **No change necessary.**

The key results presented in the main body of the paper were

1. Proportion of global enrichment split by region (Table 1)
2. Continental estimates of the area of catchments & % of world population likely to have unacceptable/acceptable growth of periphyton

These results are presented in both tabular and graphical form. **No change necessary.**

Validity

As a non-professional statistician my assessment is that the approach used was valid and well executed. The methods section and the supplementary material provide adequate information to support the conclusions. Having attempted analysis of large scale data sets I appreciate the difficulties involved in data cleaning. I was not able to assess this aspect of the approach as details were only available in a referenced publication, but I have no reason to believe that this will not have been well executed. **No change necessary.**

Significance

There is clearly a very substantial literature describing the impacts of nutrient enrichment on the water environment and thus it is not surprising that the paper concludes that anthropogenic enrichment affects nearly 50% of the world's population. In that sense there is nothing particularly original in this paper's findings. However, it does present an interesting and in my opinion analysis of a very large global data set. **No change necessary.**

The paper starts with a good, although not particularly novel, overview of the concept of reference conditions. This is important as the authors use deviation from reference conditions to assess impact. It is this approach that I feel makes the paper worthy of publication. **No change necessary. We agree that this is the first step, but some consider eutrophication in streams unimportant. However, there are negative impacts caused by stream eutrophication. To our knowledge, this is the first global analysis to attempt assessing one cause of those global impacts.**

Improvements

Key Results

In my opinion the most important results in this paper were the predictions of reference conditions derived from the models. However, these were not the main focus of the paper and I suggest that this needs to be addressed in a significant modification to the paper. Thanks, we agree. **We have taken this point on board and expanded the section on the prediction of reference conditions.**

I feel that the paper needs to include more consideration of the reference condition predictions in the main body of the paper. Currently all of the detailed information is only within the supplementary material. My suggestion is that it is this aspect of the work that is novel and needs to be brought into the main body. Figure S13 provides much useful information and would warrant further discussion. There is currently much uncertainty over the variation of reference conditions, perhaps supported but another of the authors findings in their analysis of published values (i.e. the significance of country and method), also the importance of latitude in the main models. I would like

to see more of this in the main paper. We have moved a significant proportion of the Supplementary Information into the main text and included Supplementary Figure S13 as one of the Extended data figures. We have left the discussion of caveats for reference condition performance in the Supplementary Information (referring to it) to keep the main text focused. We also included a comment about the importance of latitude in the model, which we suspect to be acting to show a seasonal effect (i.e. it was used to exclude data from colder months) or as an amalgam of other biophysical factors. We have made mention of this in the main text and the Supplementary Information.

Summary

In overview I found the paper interesting and well executed. However, in its current form I don't feel it provides sufficient new information to warrant publication, although it has conducted a worthwhile analysis of a very substantial data set. If the paper were to focus more on the issue of reference conditions in the context of geographic variability and their importance in impact assessment I do feel the paper could be worthy of publication. Point taken. In addition to 'beefing up' the reference condition section we've also made additional mention of the impact of the work for policy. However, one of the problems is that eutrophication of streams has not been considered an important problem by some in the past. Most of the focus had been on lakes. The references in this paper support the idea that eutrophication is a potential problem in rivers and streams, and we hope that this puts the problem in global perspective in addition to the potential contribution to lake and marine eutrophication.

Other comments

- Line 84 Sub-title. I did not find this sub-title particularly helpful, it needs to be modified. We changed it to "Controls on nitrogen and phosphorus losses under current and reference conditions". Hopefully, this is more relevant to the following text.
- Line 128 enrichment varied from 3% to 86% (is it not 2% for TP Table 1). Well spotted! We also noted that the data in Table 1 was listed as a proportion and a percentage in the text. WE have changed this to percentage for consistency
- Line 338 I found the legend to table 1 difficult to understand and consequently was not entirely clear what the table contained. Are the values proportions of catchments or the median enrichment level $(O-E)/E$ derived from the predicted reference values. Yes, you have interpreted the calculation correctly. We've adjusted the table heading to make that clearer.

Reviewer #3 (Remarks to the Author)

I found this paper very impressive. The most noteworthy results for me were how low both nitrogen-nitrate and total phosphorus need to be to avoid unacceptable periphyton concentrations. I think particularly the low nitrate is indicative of the prevalence of N fixing periphyton, requiring much lower concentrations in nitrate. The low concentrations of phosphorus are likely unattainable for most urbanised catchments but this paper at least provides evidence for the requirements of lower P concentrations and how much work there is left to be done. To have modelled across the globe further increases the noteworthy-ness of the work and the high impact. No change necessary.

The work is wide ranging because of the variety of fields that feed into the human impact. The combined influence of urban pressures and intensification of agriculture with the various subfactors was an interesting approach as often the human effect is segregated into rural versus non-rural pressures. It might have been useful since so much work has been done to make a stab at the impact

of those sub categories to give more clarity of what pressures need to be addressed. I think fields that are related to the sub categories, for example Environmental Planners, might feel that they need more insight into the proportion that population density is impacting the nutrient enrichment, for example. As the developed world puts limits on housing development in sensitive areas, this work could provide real evidence to support those often unpopular decisions that have been taken in catchments across the UK for example. **We understand that this would be useful. However, our analysis cannot do this because we focused on a combined human effect variable. We had thought of using each of the factors making up human effects separately but couldn't reconcile that their collinearity would likely mask the importance of any sub-factor. However, you can get a 'flavour' of the importance of each sub-factor by inspecting the partial least squares coefficients and have inserted some text to this effect.**

The discussion section was relatively short. I would suggest this is the weakest part of the paper and I think some more thinking could be done about the implications of the work that has been completed. The feat of processing 1.2 M data and making a global assessment of nutrient enrichment is rather exciting. The potential for using these findings are wide ranging - some more of that impact would be great to see in the paper.

There were a couple of areas that I either just didn't get or think could be improved. In the list of describing the 6 types of catchments, in description 4 you say if "both" conditions are met but then you say "or" when listing the conditions. I would have thought this should be "and". **Actually, after re-reading the methods – class 4 should have said co-limited and unacceptable. We've now corrected that.**

I really like how much care was taken with training datasets, testing sets, checking residuals, looking at patterns. Often this much care is missing from papers and leads to weakness in the modelling. Having read all the supplemental info and the modelling approach, I have confidence in your methods and the care you've taken. One thing I was wondering is whether it's possible to get an idea of how confident you are in your values for enrichment. You have stated values like the global-average enrichment value for nitrate was 21% but this really doesn't mean a lot to me without confidence statements. If your confidence range is something like 15-28% then that's quite different to 5-38%. **Good point. We have now included confidence intervals in Table 1. We also now mention in the Methods how these were derived (bootstrapping and resampling with replacement of 1000 datapoints).**

Also in Table S9, which nutrient are you referring to in column 5? **We have changed this to Analyte median or concentration.**

I'd also expect an comment on some clear outliers in the comparison between estimated and actual values (Fig S11). I appreciate they're both in the supplemental data but I haven't found mention of any particular outliers in the main body text - only mention of catchments that were removed due to a lack of water. **WE have re-inspected these data. They are outliers, but we've chosen not to remove them owing to their small number (<1% of the total N pool) and their close alignment to the 1:1 line. There is also some literature to support keeping low (but not high) outliers in datasets – owing to the fact that we know low concentrations are low, but high concentrations could be an 'out of range error'.**

Overall though I found the description of methods and the process excellent - and if this paper is published then something I'd be able to teach my students. I think the main area to improve on is the significance of the results and how they impact the topics chosen for discussion. Perhaps widening the discussion a little to describe that impact will achieve this. **Thanks, no change required. However, we have expanded the discussion a bit, but did not want to overburden the manuscript with too much speculation. We hope that the level of consideration now is consistent with your comments.**